Predicting energy use in construction using Extreme Gradient Boosting

http://orcid.org/0009-0009-8857-5725 Han Jiaming 1 jiaming.han@connect.polyu.hk
Shu Kunxin 1
Wang Zhenyu 2
1 Department of Computing, The Hong Kong Polytechnic University , Hong Kong , Hong Kong SAR
2 School of Mechanical Engineering, Hefei University of Technology , Anhui , China
Nguyen Binh
Electronic publication date: 2023 Aug 7
Publication date: 2023
Volume: 9
Electronic Location ID: e1500
Received 2023 Apr 26; Accepted 2023 Jul 4
Copyright: © 2023 Han et al.
Copyright year: 2023
Copyright holder: Han et al.
License: This is an open access article distributed under the terms of the Creative Commons Attribution License, which permits unrestricted use, distribution, reproduction and adaptation in any medium and for any purpose provided that it is properly attributed. For attribution, the original author(s), title, publication source (PeerJ Computer Science) and either DOI or URL of the article must be cited.
License URL: https://creativecommons.org/licenses/by/4.0/

Keywords: Artificial intelligence, Data mining and machine learning, Data science, Prediction, Gradient boosting, Energy, Time-series

Funding: The authors received no funding for this work.

==============================
Annual increases in global energy consumption are an unavoidable consequence of a growing global economy and population. Among different sectors, the construction industry consumes an average of 20.1% of the world’s total energy. Therefore, exploring methods for estimating the amount of energy used is critical. There are several approaches that have been developed to address this issue. The proposed methods are expected to contribute to energy savings as well as reduce the risks of global warming. There are diverse types of computational approaches to predicting energy use. These existing approaches belong to the statistics-based, engineering-based, and machine learning-based categories. Machine learning-based frameworks showed better performance compared to these other approaches. In our study, we proposed using Extreme Gradient Boosting (XGB), a tree-based ensemble learning algorithm, to tackle the issue. We used a dataset containing energy consumption hourly recorded in an office building in Shanghai, China, from January 1, 2015, to December 31, 2016. The experimental results demonstrated that the XGB model developed using both historical and date features worked better than those developed using only one type of feature. The best-performing model achieved RMSE and MAPE values of 109.00 and 0.24, respectively.

Introduction

The annual rise in global energy use is one of the direct consequences of economic and population expansion. Energy use in the construction sector, which accounts for an average of 20.1% of the world’s total energy use, is a vital aspect of the global energy consumption picture (Conti et al., 2016). In many nations, this proportion is substantially greater; for example, in China and the United States, it contributes up to 21.7% and 38.9% of total energy use, respectively (China Association of Building Energy Efficiency, 2021; Becerik-Gerber et al., 2014). This rising energy use worsens global warming and causes natural resource depletion. Therefore, increasing the efficient use of energy in the construction sector is essential because it lessens the risk of global warming and enhances sustainable growth. To effectively manage diverse technical activities, e.g., demand response during construction (Dawood, 2019), organization of urban power systems (Moghadam et al., 2017), and defect detection (Kim et al., 2019), prediction of energy demand is crucial in preventing ineffective energy use in many sectors (Min et al., 2023; Li et al., 2022), especially construction (Liu et al., 2023; Zhang, Li & Yang, 2021). Besides, a good estimation of energy demand helps to create energy-saving plans for heating, ventilation, and air conditioning (HVAC) systems (Du et al., 2021; Min, Chen & Yang, 2019). To address this issue, several computation frameworks were proposed to predict energy use. Zhao & Magoulès (2012) performed a comprehensive review of numerous computational frameworks developed to estimate construction energy use. According to their investigation, all approaches for predicting energy use may be generally categorized into three groups: statistics-based, engineering-based, and machine learning-based techniques.

Statistics-based approaches were employed to model the correlation between energy use and involved attributes with a parametrically defined mathematical formula. Ma et al. (2010) developed models using multiple linear regression and self-regression techniques and essential energy consumption features characterized by particular population activities and weather conditions. The least squares method was used to estimate parameters as well as approximate monthly energy consumption for broad-scale public housing facilities (Ma et al., 2010). Conducting the principal component analysis, Lam et al. (2010) discovered three significant climatic variables, comprising dry-bulb temperature, wet-bulb temperature, and global solar radiation, to create a novel climatic index. After that, regression models were built to establish an association focusing on the daily period between the climatic index and the simulated cooling demand. Despite fast and simple calculations, these approaches often lack flexibility and have poor prediction ability due to limitations in handling stochastic occupant behaviors and complicated factor interactions between factors (Ahmad et al., 2014).

Applying physics’ concepts and thermodynamic calculations, engineering-based approaches determine the energy use of each building component (Zhao & Magoulès, 2012). Although the defined relationships between input and output variables improve the model’s explainability, these approaches need curated sources of building and environmental data whose accessibility is very limited (Zhao & Magoulès, 2012). Some researchers have attempted to reduce the complexity of engineering models in order to improve their ability to accurately predict energy use. Yao & Steemers (2005) developed a straightforward approach for generating a load profile (SMLP) to predict the breakdown of everyday energy consumed by electrical household appliances. This technique estimated seasonal energy demand since the average daily energy (consumed by these appliances) seasonally fluctuated. Based on frequency response characteristic analysis, Wang & Xu (2006) obtained simple physical properties to determine the model parameters. In addition, they used monitored operating data to identify parameters for a thermal network of lumped thermal mass to describe a building’s interior mass (Wang & Xu, 2006). The prediction accuracy, however, is one of the most concerning issues because the simplified model may be somewhat underfitting (Wei et al., 2018).

Machine learning-based approaches utilize either traditional machine learning (e.g., random forest, support vector machines) or deep learning (e.g., artificial neural networks (ANN), convolutional neural networks (CNN), recurrent neural networks (RNN)) for modeling by learning from historical data (Nora & El-Gohary, 2018; Tian et al., 2019). These machine learning-based approaches usually exhibit better performance compared to other methods, especially in event detection (Wang et al., 2023; Sun et al., 2023c; Ren et al., 2022) and other applications using time-series data (Sun et al., 2023a, 2023b; Long et al., 2023). To predict perennial energy use, Azadeh, Ghaderi & Sohrabkhani (2008) suggested using ANN in combination with an analysis of variance. The approach was demonstrated to outperform the standard regression model. Hou & Lian (2009) developed a computational framework using support vector machines to estimate the cooling demand of an HVAC system, and their findings showed that it was dominant over autoregressive integrated moving average (ARIMA) models. Tso & Yau (2007) used decision tree (DT), stepwise regression, and ANN to estimate Hong Kong’s power consumption. The results suggested that the DT and ANN techniques performed somewhat better during the summer and winter seasons, respectively. However, many traditional machine learning approaches use shallow frameworks for modeling, reducing prediction efficiency. As black box models, deep neural networks do not well explain physical behaviors, but they can effectively learn abstract features from raw inputs to build superior models (Ozcan et al., 2021). Cai, Pipattanasomporn & Rahman (2019) constructed prediction frameworks using RNN and CNN to measure time-series building-level load in recursive and direct multi-step approaches, respectively. Compared to the seasonal ARIMA model with exogenous inputs, the gated 24-h CNN model had its prediction accuracy improved by 22.6% in the experiments. To calculate electric load consumption, Ozcan, Catal & Kasif (2021) suggested dual-stage attention-based recurrent neural networks consisting of encoders and decoders. Experimental results revealed that their proposed method vanquished other methods (Ozcan, Catal & Kasif, 2021). A hybrid sub-category of deep learning called deep reinforcement learning (DRL) blends reinforcement learning decision-making with neural networks (Mnih et al., 2015; Levine et al., 2016; Sallab et al., 2017). DRL approaches are often used in the construction industry to investigate the ideal HVAC control (Wei, Wang & Zhu, 2017; Huang et al., 2020). DRL approaches obtained promising outcomes for energy use prediction (Wei, Wang & Zhu, 2017; Huang et al., 2020). Liu et al. (2020) examined the efficacy of DRL approaches for estimating energy consumption, and their results suggested that the deep deterministic policy gradient (DDPG) method had the most predictive power for single-step-ahead prediction. Although the deployment of DRL approaches yields initial success, further research is required to determine its ability to solve this issue.

In this study, we proposed a more effective method to predict energy use in the construction sector using four tree-based machine learning algorithms, including random forest (RF), extremely randomized trees (ERT), extreme gradient boosting (XGB), and gradient boosting (GB). Four feature sets are created to combine with four learning algorithms to form 16 models. The four feature sets correspond to four development and testing scenarios: (1) using 1–3 historical hours, (2) using 1–5 historical hours, (3) using 1–7 historical hours, and (4) using 1–9 historical hours before prediction time as input features. The best-performing models are then selected as the final models for evaluation.

Materials and Methods

Dataset

In this study, we used an energy consumption dataset recorded in an office building in Shanghai (Fu et al., 2022). The energy consumption data were recorded hourly from January 1, 2015, to December 31, 2016. Since data from February 29, 2016 was not recorded, the dataset has a total of 17,520 recorded samples. The original dataset was split into three parts: ‘From 1 January 2015 to 30 April 2016’, ‘From 1 May 2016 to 31 August 2016’, and ‘From 1 September 2016 to 31 December 2016’ which were used as a training set, validation set, and test set. In the case of time series data, random samples might lead to a failed training session since the model could not learn the data that were included in the validation or test set. The information about the data sets is summarized in Table 1.

Table 1 Data for model development and evaluation.

Dataset	No. of samples	Time period	
Training	13,872	From 3 January 2015 to 3 August 2016	
Validation	1,440	From 4 August 2016 to 2 October 2016	
Test	2,160	From 3 October 2016 to 31 December 2016	
Total	17,472	From 3 January 2015 to 31 December 2016	

Overview of method

Figure 1 provides a flowchart of the modeling strategies of our work with two stages: validation and testing. The validation stage aims to select the best model, while the testing stage focuses on evaluating the performance of the selected one. The modeling steps of both stages are similarly designed. As the samples are time-series, training, validation, and test data are partitioned in time-sequential order. The validation and test sets contain 1,440 and 2,160 samples, which correspond to 60 and 90 days, respectively. Initially, the model is developed with the training data and then evaluated on a timely ordered list of seven sub-validation sets before being retrained. Each sub-validation set contains 336 validation samples, which are equivalent to 14 days. After 7 days, the model is retrained with an updated training set, which is a combination of previous training samples and 7-day validation samples. This process is repeated until no validation samples are left. After completing the validation stage, we compare the performance of these models to select the best one. The best model is then retrained with the new training set, which contains all training and validation samples. The obtained model is finally evaluated on the test data in the same manner as in the validation stages.

Figure 1 Overview of method.

Data processing and featurization

Figure 2 describes the main steps in data sampling, processing, and featurization. After obtaining the training, validation, and test sets, we processed the data to generate their feature sets. The original data frame has only two columns: ‘recorded time’ and ‘energy used’. We created two types of features, including ‘historical feature’ (Fig. 2B) and ‘date feature’ (Fig. 2C). The historical features are energy values that were recorded 1,2,...,n h before the prediction time with n=48 (2 days). The label i (value of energy used) of sample i is filled to the column ordered 48th of sample i+1. The columns ordered 48th,47th,46th,...,1st present the recorded values of energy used before 1,2,3,...,48 h, respectively. The first sample was recorded on ‘January 1, 2015 at 00 h’ with undetermined historical features. The next 47 samples have their historical features determined with 1,2,3,...,47 recorded values. Since the first 48 samples have historical features containing undetermined values, we removed them from the dataset. The samples recorded after ‘January 3, 2015 at 00 h’ have the historical features of all determined values. The date features of a sample give information on ‘day of year’, ‘whether it is a holiday’, and ‘when it is recorded’. Gaining information on whether a day is a holiday because the energy use in holiday time will be significantly different from daily energy use. Besides international holidays, Eastern countries like China celebrate some holidays based on the lunar calendar (e.g., Chinese New Year). The historical and date features are expressed as 48-dimensional and three-dimensional numeric vectors, respectively. To select the best feature set, we examined three scenarios, including using only historical features, using only date features, and using combinatory features (Table 2).

Figure 2 Major steps in data sampling, processing, and featurization.

(A) Data sampling, (B) creating historical variables, (C) extracting binary timing vectors.

Table 2 Scenarios for model development and evaluation.

Scenario	No. of features	Annotation	
1	48	Using 1, 2,…, 48 h before prediction time as historical features	
2	3	Using day of year, holiday information, and recorded hour as date features	
3	51	Using both historical features and date features as combinatory features	

Machine learning algorithms

Random forest

Random forest (RF) (Breiman, 2001) is a supervised machine learning method that utilizes an ensemble of decision trees. It was developed based on the “bagging” concept (Breiman, 1996) and the use of random feature selection (Ho, 1995), which allows for the creation of multiple decision trees that are distinct from one another. In RF, the output is determined by either the mode of the classes or the average of the predicted values of the multiple trees, depending on the task at hand (i.e., classification or regression). One of the key benefits of RF is its ability to overcome the issue of overfitting, which is a common problem in decision tree algorithms. Overfitting occurs when a model becomes too closely fitted to the training data, leading to poor generalization to unseen data. RF is able to mitigate this issue due to its use of multiple decision trees and the random feature selection process.

Extremely randomized trees

Extremely randomized trees (ERT) (Geurts, Ernst & Wehenkel, 2006) is a supervised machine learning method that is designed to address both classification and regression tasks. It is based on the principles of tree-based ensembles, and the central aspect of this method is the significant randomization of both the attributes and the cut points used to divide a tree node. In some cases, ERT may construct trees whose structures are entirely random and unrelated to the output values of the training sample. The degree of randomization can be controlled through the use of a parameter, allowing the user to adjust the intensity of randomization as needed. ERT is known for its accuracy as well as its improved computational speed compared to other tree-based methods such as random forest.

Extreme gradient boosting

eXtreme Gradient Boosting (XGB) (Chen & Guestrin, 2016) is a powerful supervised machine learning algorithm that combines the principles of gradient tree boosting (Friedman, 2001; Mason et al., 2000) with classification and regression trees (CART) (Steinberg & Colla, 2009). It is equipped with additional regularization options, including L1 and L2 penalization. XGB is trained to minimize a regularized objective function that includes a convex loss function and a penalty scoring function based on the difference between the predicted outcomes and the true labels. Its boosting strategy involves the successive use of random subsets of data and features, with the weight of mispredicted classes increasing in each iteration. This method has been shown to be effective for a range of tasks.

Gradient boosting

Gradient boosting (GB) (Friedman, 2001, 2002) is a supervised machine learning technique that is based on the concept of improving the performance of a weak learner through the use of an ensemble approach. It involves sequentially training weak learners on filtered subsets of the data, with each successive learner focusing on the observations that were not well predicted by the previous learners. GB can be applied to a variety of tasks, including regression, multi-class classification, and other problems, by utilizing arbitrary differentiable loss functions. It is a flexible method that can be generalized to a wide range of situations.

Model selection

To select the best-performing model for testing, we validated 12 models (created by four learning algorithms in combination with three feature sets). Since all models were developed using tree-based algorithms, data normalization can be bypassed. The parameter max_depth was used for model tuning, while the other parameters were set to default once the best model was selected. The model was selected based on mean absolute percentage error (MAPE) and root mean squared error (RMSE).

Results and discussion

The model’s performance was evaluated using two metrics: root mean squared error (RMSE) and mean absolute percentage error (MAPE).

(1) RMSE=∑i=1N(y^i−yi)2n,

(2) MAPE=1n∑i=1N|Ai−FiAi|,

where n is the number of samples, y^ is the predicted value, and y is the true value.

Model validation

Table 3 provides a comparative analysis of the performance of the 12 developed models on the validation set. Generally, models developed using combinatory features obtain better performances, followed by those developed with historical features and those developed with date features only. The GB model shows better performance than other models in scenario 2, with the smallest RMSE and MAPE values. While in scenarios 1 and 3, the XGB models work more effectively compared to the others, the ERT models have limited performance in all scenarios in terms of RMSE and MAPE. The GB models are ranked as the second-best models in scenarios 2 and 3. The performance of RF models is not competitive in all scenarios. The results are not surprising because the RF and ERT algorithms share more common characteristics, while the XGB algorithm is developed based on the concept of the GB algorithm. Based on the results, we selected the XGB model developed using combinatory features as our final model for evaluation (Table 4).

Table 3 Performance of the 12 models on the validation set.

Scenario	Model	Metric	
RMSE	MAPE	
1	RF	227.76	0.50	
	ERT	364.06	0.96	
	XGB	238.02	0.47	
	GB	264.72	0.57	
2	RF	311.39	0.78	
	ERT	322.48	0.70	
	XGB	309.35	0.72	
	GB	277.25	0.53	
3	RF	245.76	0.28	
	ERT	186.54	0.37	
	XGB	133.56	0.21	
	GB	150.41	0.25	

Table 4 Performance of the tuned XGB models on the validation set.

Parameter (max_depth)	Metric	
RMSE	MAPE	
2	146.87	0.32	
3	112.38	0.20	
4	124.3	0.21	
5	135.29	0.21	

Model evaluation

Figure 3 gives information on the variation of prediction error over multiple time points of test data. The results indicate that the prediction errors show downward trends over the testing period. The errors tend to increase when the time point’s distance to the retraining point is large and then immediately drop once we retrain the model with updated training data. Figures 3A and 3B visualize the changes in RMSE and MAPE over 76 time points, where each time point represents the 14-day period with between-point distances of 1 day. Besides, Figures 3C and 3D illustrate the performance of the best and worst predicted time points over 336 consecutive hours. To quantify the variation of prediction error, we computed the mean, standard deviation, and 95% confidence interval (95% CI). The RMSE achieves a mean of 109.00 and a standard deviation of 37.07, while the MAPE obtains a mean of 0.24 and a standard deviation of 0.08. The 95% CI values of RMSE and MAPE are [100.47–117.52] and [0.22–0.26], respectively (Table 5).

Figure 3 Variation in prediction error over multiple time points of test data.

(A) MAPE, (B) RMSE, (C) best predicted time point, (D) worst predicted time point.

Table 5 The statistical values of metrics on the test set.

Metric	Statistics	
Mean	Standard deviation	95% Confidence Interval	
RMSE	109.00	37.07	[100.47–117.52]	
MAPE	0.24	0.08	[0.22–0.26]	

Conclusions

The experimental results confirm the robustness and effectiveness of machine learning, especially Extreme Gradient Boosting (XGB), in developing computational models predicting energy use. The achieved results indicate that the XGB models work more effectively than other tree-based models to address this issue. Also, the selection of a suitable historical period is essential to improving model performance.

Supplemental Information

Supplemental Information 1 Code and data.

Click here for additional data file.

Additional Information and Declarations

Competing Interests

Author Contributions

Data Availability

The authors declare that they have no competing interests.

Jiaming Han conceived and designed the experiments, performed the experiments, analyzed the data, performed the computation work, prepared figures and/or tables, authored or reviewed drafts of the article, and approved the final draft.

Kunxin Shu performed the experiments, analyzed the data, performed the computation work, prepared figures and/or tables, authored or reviewed drafts of the article, and approved the final draft.

Zhenyu Wang analyzed the data, performed the computation work, prepared figures and/or tables, authored or reviewed drafts of the article, and approved the final draft.

The following information was supplied regarding data availability:

The code and data are available in the Supplemental File.

The data used in this study is from Fu et al. (2022) and is available at GitHub: https://github.com/gltzlike/DF-DQN-for-energy-consumption-prediction/tree/master/data (accessed on 10 March 2023).

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
