# Peer review of "Predicting energy use in construction using Extreme Gradient Boosting"

_PeerJ Computer Science, doi:10.7717/peerj-cs.1500_

## Round 0.1 · original submission · Major Revisions

All of the reviewers recommended that additional experiments be conducted, with reviewers 2 and 3 recommending that the model testing procedure be enhanced. The reviewers suggested using statistical analysis to report confidence intervals for the model's performance. It is essential to address these recommendations, as doing so would provide additional insight into the robustness and validity of the proposed model.

Reviewer 1 ·

Basic reporting

The manuscript was well-written with clear and unambiguous technical language. The introduction and background sections provide sufficient context for the study, and the relevant prior literature has been appropriately referenced throughout the manuscript. The figures are of high quality, in vector format, and are nicely displayed in the manuscript. They are relevant to the content of the manuscript and appropriately described and labeled.

The raw data was shared via a Github link, which allows for easy accessibility and reproducibility. The manuscript is 'self-contained' and includes all relevant results, and it clearly fits the aims and scope of the journal.

Experimental design

The research question is defined clearly and the study makes a valuable contribution to filling a knowledge gap. Although the methods are relatively simple, they were described clearly in the manuscript, and the study is reproducible as the code was provided. While the study may not be particularly novel, it is beneficial to the literature as the software is reproducible and shows good results.

Validity of the findings

The authors have provided all the necessary data, although additional experiments could be conducted to further demonstrate the statistical soundness of the results. The conclusions were appropriately stated, connected to the original question investigated, and limited to those supported by the results.

Additional comments

To improve the manuscript, the authors should address the following points:
+ Summarize the context of the experiments, including details about the dataset and experimental settings, in the abstract.
+ Fix some references in the text to match those in brackets, such as changing "Hou et al." to "Hou and Lian", "Tso" to "Tso and Yau", and "Wang et al." to "Wang and Xu".
+ Add the missing citation for "Zhao and Magoul".
+ Include mean absolute percentage error (MAPE) in addition to RMSE as an evaluation metric, as it is easy to understand.
+ Tune the models rather than setting only one hyperparameter (i.e., max_depth), since there is a validation set available.

Cite this review as

Reviewer 2 ·

Basic reporting

The manuscript is well-written in professional English and well-organized. The literature review is sufficient and clear. The results are clearly described in tables and figures. The results efficiently support the results but there are a lot of issues that need to be addressed before being considered for publication in this journal.

Experimental design

The design is correct but the testing process is too simple. Since the data is time-series data, the testing process needs to be re-designed to meet the nature of the dataset.
(1) Authors are suggested to create a number of sub-test sets (derived from the original test set) in which each sub-test set represents a different time point.
(2) The research question is well-defined and addressed.
(3) The reproducibility of the work should be questioned. The authors need to provide evidence of statistics basis (e.g., 95%CI) and graphs (e.g., histogram) to indicate that the work is reproducible.
(4) The methodology is clear with simple data encoding. However, the authors used 4 conventional machine learning algorithms only. Did authors try other algorithms?

Validity of the findings

The findings are valuable. Although the methods are not novel, it's still fine if they can address the research question. However, experimental replication needs to be conducted to support the conclusion. The dataset is accessible and the source code is clean. The conclusion is consistent with the objective. However, I think the "Conclusions" should be "Conclusion" because authors draw only one conclusion to answer their only objective.

Cite this review as

Reviewer 3 ·

Basic reporting

The work under review appears to be well-structured and written in a formal style of English. The literature review is adequate and provides a comprehensive overview of the subject matter. The tables and figures included in the manuscript are well-designed and present the research findings in a concise and clear manner. However, despite these positive aspects, the manuscript still needs some improvements before it can be considered ready for publication. Specifically, there are several issues listed below that need to be clarified.

Experimental design

The study in question has effectively answered the research question at hand, however, it appears that the testing process may require improvement. Specifically, the authors have utilized the entire test set for a single round of testing. While this approach may not be technically wrong, it is not considered practical. As a solution, it is suggested that the authors modify their problem statement to predict a specific duration, such as two weeks or a month in advance, rather than attempting to predict the entire test set in one go. This not only allows for a more practical application of the model, but also enables a statistical analysis of the model's performance, providing greater insight into the effectiveness of the approach.

Validity of the findings

The findings of the study can be valuable for researchers in various fields who are interested in similar topics. The dataset and code repository are made publicly available, which makes it easier for others to replicate the results and extend the study. However, it is important to note that to confirm the robustness of the proposed model and the validity of the findings, certain modifications to the experiments may be necessary. It is recommended that future studies consider using alternative evaluation metrics and changing the experiments as suggested above. By doing so, the results of the study can be further validated and potentially lead to more accurate and effective solutions to problems in machine learning.

Cite this review as

---

## Round 0.2 · Minor Revisions

Please address the comments from Reviewer 2 and revise the manuscript accordingly.

Reviewer 1 ·

Basic reporting

The manuscript employs clear and unambiguous technical terminology, as well as extensive referencing of literature. The figures are in good quality and vector format.

Experimental design

The authors have significantly improved the experimental design, making it more practical and valuable. The provision of code and data ensures reproducibility of the study. Although the technique may not be novel, it offers practical benefits to the relevant communities, especially in the context of common time-series prediction tasks.

Validity of the findings

Additional experiments have been conducted to further demonstrate the statistical soundness of the results. A new evaluation metric, MAPE, has been included to make the results easier to understand. The conclusions were appropriately stated, connected to the original question investigated, and limited to those supported by the results.

Additional comments

The authors have addressed all of my concerns. The manuscript has been much improved. It can be accepted for publication.

Cite this review as

Reviewer 2 ·

Basic reporting

The manuscript has significantly revised. The language used is fine except some details:
- Tablle 1: "From 1 January 2015 to 31 December 2016" --> It shoud be from "From 3 January 2015 to 31 December 2016"

Experimental design

The experimental design is clear and detailed. This revised version is significantly improved compared to the first submissions with many details added to emphasise the contribution of authors. However, are some minor questions that need to be explained.

- Is there any reason to replace R2 metric with MAPE? Why don't you keep it?
- You changed the new encoding method for time-series data to focus on holidays of years. It makes sense but authors need to add more details to clarify it for readers.

Validity of the findings

Authors said that they evaluate the models over 76 time points points, where each time point represents the 14-day period. It is a good approach, especially much better than the previous version. The findings now support the validation of the study with statistic evidence.

Cite this review as

Reviewer 3 ·

Basic reporting

The work under review appears to be well-structured and adheres to a formal style of English. The literature review adequately covers the subject matter, offering a comprehensive overview. The manuscript includes well-designed tables and figures that effectively present the research findings with clarity and conciseness.

Experimental design

The authors have implemented a significant revision in the manuscript by modifying their models to predict a specific duration in advance instead of attempting to predict the entire test set simultaneously. This adjustment offers a more practical application of the model. Moreover, the revised manuscript includes a statistical analysis of the model's performance, which provides valuable insights into the effectiveness of the approach.

Validity of the findings

The findings seem to be valid and can be useful for researchers from different fields who are interested in time-series prediction or similar topics. The dataset and code repository associated with the study are publicly accessible, making it easier for others to reproduce the results and build upon the study.

Additional comments

The revised version of the manuscript is much better than the first version. It can be accepted for publication.

Cite this review as

---

## Round 0.3 · accepted · Accept

The authors have addressed all of the reviewers' comments. I have assessed the revision and, in my opinion, the manuscript is ready for publication.